# Combinations of Cannabidiol and Δ^9^-Tetrahydrocannabinol in Reducing Chemotherapeutic Induced Neuropathic Pain

**DOI:** 10.3390/biomedicines10102548

**Published:** 2022-10-12

**Authors:** Diana E. Sepulveda, Kent E. Vrana, Nicholas M. Graziane, Wesley M. Raup-Konsavage

**Affiliations:** 1Department of Pharmacology, Penn State College of Medicine, Hershey, PA 17033, USA; 2Department of Anesthesiology & Perioperative Medicine, Penn State College of Medicine, Hershey, PA 17033, USA

**Keywords:** neuropathic pain, tetrahydrocannabinol, cannabidiol, cannabinoids, cannabis

## Abstract

Neuropathic pain is a condition that impacts a substantial portion of the population and is expected to affect a larger percentage in the future. This type of pain is poorly managed by current therapies, including opioids and NSAIDS, and novel approaches are needed. We used a cisplatin-induced model of neuropathic pain in mice to assess the effects of the cannabinoids THC and CBD alone or in varying ratios as anti-nociceptive agents. In addition to testing pure compounds, we also tested extracts containing high THC or CBD at the same ratios. We found that pure CBD had little impact on mechanical hypersensitivity, whereas THC reduced mechanical hypersensitivity in both male and female mice (as has been reported in the literature). Interestingly, we found that high CBD cannabis extract, at the same CBD dose as pure CBD, was able to reduce mechanical hypersensitivity, although not to the same level as high THC extract. These data suggest that, at least for CBD-dominant cannabis extracts, there is an increase in the anti-nociceptive activity that may be attributed to other constitutes of the plant. We also found that high THC extract or pure THC is the most efficacious treatment for reducing neuropathic pain in this model.

## 1. Introduction

Neuropathic pain is a condition that is caused by damage to the nervous system as a result of physical trauma, chemotherapy, infection, metabolic disease, or autoimmune disorders [1,2]. This chronic pain condition affects 7–10% of the population and, unfortunately, current drug treatments have poor efficacy and tolerability [3,4]. Patients are typically unresponsive to analgesics and opioids. In addition to pain, patients also suffer from sleep disturbances, anxiety and depression, and reduced quality of life [5]. The incidence of neuropathic pain is expected to increase in the coming years due in part to the diabetes epidemic, improved cancer survival rates, and age [6]. Therefore, new approaches to treat pain in these patients are clearly needed.

Cannabis has been used for centuries to treat pain, and the plant contains a number of pharmacologically active compounds including cannabinoids and terpenes that might have anti-nociceptive properties [7,8]. The two most abundant and studied cannabinoids in the plant are Δ^9^-tetrahydrocannabinol (THC) and cannabidiol (CBD), and the ratio of these two compounds can vary greatly between cultivars and subtypes of *Cannabis*. Both THC and CBD have been shown to reduce neuropathic pain in animal models [9,10,11,12,13]. However, patients rarely take pure THC or CBD for pain. Additionally, THC and CBD are normally administered together at varying ratios depending on the product consumed. Furthermore, the anti-nociceptive effects of these varying ratios of THC and CBD remain underexplored. Instead, studies have largely focused on THC:CBD co-administered at 1:1 ratios, commonly found in the European-approved Sativex (nabiximols) that has been shown to be effective at treating pain in patients with certain conditions [14,15,16,17]. We therefore set out to assess how different combinations of THC and CBD, as both pure compounds and from unfractionated plant extracts, affect mechanical hypersensitivity in a mouse model of chemotherapeutic-induced peripheral neuropathy (CIPN).

## 2. Materials and Methods

### 2.1. Animals

All experiments were conducted in a manner approved by the Pennsylvania State University, College of Medicine Institutional Animal Care and Use Committee (IACUC). Male (*n* = 120) and female (*n* = 120) age-matched (10–12 weeks) wild-type C57BL/6 mice (The Jackson Laboratory, Bar Harbor, ME) were used in this study. All mice were group-housed with a 12 h light/dark cycle with access to food and water ad libitum.

### 2.2. Drugs

Cannabidiol (CBD) and Δ^9^-tetrahydrocannabinol (THC) were purchased from Cayman Chemical (Ann Arbor, MI, USA). CBD extract was produced through supercritical CO_2_ extraction from hemp (Helping Hands Hemp, Womelsdorf, PA, cultivar: YoungSim 10) and THC extract was provided by the NIDA drug program (Research Triangle Institute, Research Triangle Park, NC, USA) the composition of both was verified by an independent laboratory (Keystone State Testing, Harrisburg, PA, USA) Cisplatin was purchased from Acros Organics (Fairlawn, NJ, USA).

### 2.3. Supercritical CO_2_ Extraction

Dried hemp flower, 500 g, was ground and then extracted using supercritical CO_2_ in an extractor from Supercritical Fluid Technologies (CannabisSFE, Newark, DE, USA). Extraction was performed at 55 °C, with 413 bars of pressure for 30 min. Extract was collected and dissolved in ethanol to a concentration of 10% extract in 90% ethanol by weight and incubated at −20 °C for 24 h (winterization). The solution was then filtered and ethanol was evaporated. The extract was then resuspended in fractionated coconut oil (Pure Body Naturals, West Chester, OH, USA) at 200 mg/mL and heated to 95 °C for 1 h to decarboxylate the cannabinoids.

### 2.4. Cisplatin-Induced Neuropathy

Peripheral neuropathy was induced by injecting mice with 5 mg/kg of cisplatin intraperitoneally (IP) once weekly for four weeks, as previously described [13,18]. The mice were co-administered 1 mL of 4% sodium bicarbonate solution subcutaneously prior to the cisplatin injection to reduce nephrotoxicity and to minimize compromised renal functions [13,18]. Mechanical allodynia was assessed using an electronic von Frey anesthesiometer equipped with a semi-flexible polypropylene super-tip (IITC Life Science Inc., Woodland Hills, CA, USA), these assessments were made before and after cisplatin treatment to confirm neuropathic pain state, as described below (Data from the pre and post cisplatin assessments are presented in Appendix A).

### 2.5. Von Frey Testing

Hypersensitivity to mechanical pressure was assessed using an electronic von Frey anesthesiometer (IITC Life Sciences Inc.). For testing, mice were placed in small acrylic chambers on a wire mesh table (IITC Life Sciences Inc.). Animals were allowed to acclimate to the chamber for 20 min prior to testing. The von Frey anesthesiometer was equipped with a semi-flex tip (IITC Life Sciences Inc.), that was applied to the plantar surface of the right hind-paw with increasing force to prompt a withdrawal response. The averages from three tests were calculated with each test being separated by a minimum of 3 min. To measure the effects of test compounds, neuropathic mice were randomly assigned to one of 6 groups and injected intraperitoneally (i.p.) with vehicle (DMSO, Tween 80, saline (1:1:18), i.p.), THC at 6 mg/kg, THC and CBD in combination at 4 mg/kg and 2 mg/kg; 3 mg/kg and 3 mg/kg; 2 mg/kg and 4 mg/kg, respectively, or CBD at 6 mg/kg 1 h prior to tests. These groups represent THC:CBD ratios of 1:0, 2:1, 1:1, 1:2, or 0:1, as shown in Table 1. All von Frey measurements were performed by experimenters blinded to treatments.

### 2.6. Statistical Analysis

All results are shown as mean ± standard deviation. Statistical significance was determined using GraphPad Prism Software (9.3.1, San Diego, CA, USA) using a one-way ANOVA with Tukey’s correction for multiple comparisons. Two-tailed tests were used for all comparisons. 

## 3. Results

### 3.1. THC, but Not CBD, Reverses Mechanical Hypersensitivity in Neuropathic Male Mice

To assess the ability of pure CBD or THC to reduce cisplatin-induced neuropathic pain, von Frey tests were conducted to measure mechanical sensitivity in neuropathic male mice treated with varying doses (5, 10, or 20 mg/kg) of pure CBD or THC. Acute CBD treatment, administered 1 h prior to von Frey testing, had no significant impact on mechanical hypersensitivity in neuropathic male mice (Figure 1A). In contrast, there was a dose dependent effect of THC on pain in these animals; with only the lowest dose (5 mg/kg) being unable to reduce pain compared to vehicle treated animals (Figure 1B). Based on these findings, in order to investigate the potential interaction between these two compounds, we used a standard dose of 6 mg/kg of total cannabinoid (THC alone, CBD alone, or a combination of various THC:CBD ratios) for all subsequent experiments, with the knowledge that, at this dose, the THC (THC:CBD ratio of 1:0) would serve as a positive control, based upon previous work [13]. The dose of 6 mg/kg was selected for two reasons, first we wanted to be able to have a dose that limited the cataleptic effects of THC on mice and, second, we wanted a dose of THC that would be responsive to any additive effects of CBD on reducing hyperalgesia.

### 3.2. CBD Does Influence THC-Induced Decreases in Mechanical Hypersensitivity

We next investigated the anti-nociceptive effects of five THC and CBD combinations while maintaining the total cannabinoid administered at 6 mg/kg as shown in Table 1. In neuropathic male mice, we observed a statistically significant decrease in mechanical hypersensitivity after administration of THC:CBD only at the 2:1 ratio (Figure 2A). In contrast, in neuropathic female animals we observed a statistically significant decrease in mechanical hypersensitivity after administration of THC:CBD at ratios of 1:2 and 2:1 (Figure 2B). While the addition of CBD reduces the concentration of THC necessary to reduce hyperalgesia (note that 5 mg/kg pure THC was without effect in Figure 1), no combination of CBD and THC is greater at reducing sensitivity than THC alone.

### 3.3. CBD and THC Extracts Are Effective at Attenuating Mechanical Hypersensitivity in Neuropathic Mice

We next compared the anti-nociceptive effects of THC and CBD ratios using CBD-dominant or THC-dominant botanical extracts. Extracts were mixed at varying ratios of THC: CBD. The composition of the extracts (as delivered to the mice) for cannabinoid and most abundant terpene content (Appendix A), are shown in Table 2 and Table 3, respectively.

For full composition of the undiluted extracts please see Appendix A. Using a CBD-dominant botanical extract (containing 40 µg/mL THC; labeled “0:1”), we found a statistical difference between vehicle and treated animals for both sexes (Figure 3A,B).

In neuropathic male mice, we saw a further reduction in mechanical hypersensitivity when THC was added (1:2, 1:1, and 2:1 ratios compared to 0:1, CBD extract), but no change in hypersensitivity as THC concentration increased (Figure 3A). In contrast, in neuropathic female mice, we did not see any further reduction in mechanical hypersensitivity when THC was included in the treatment. That is, the reduction in hypersensitivity was consistent across the tested ratios with what was observed for the CBD extract (0:1 ratio). However, the THC extract alone (1:0 ratio) was significantly better than the CBD extract (Figure 3B). Because the CBD extract contains trace amounts of THC and vice versa; it was not possible to have any extract ratio where the other cannabinoid was exactly zero. Table 4 shows the concentration of THC and CBD delivered at each of the ratios.

## 4. Discussion

In this study, we examined the effects of CBD and THC on pain associated with chemotherapeutic-induced peripheral neuropathy (CIPN), both as pure compounds and botanical extracts at varying ratios of THC:CBD. These data are important because patients taking medical cannabis or cannabis-based products do not typically take pure CBD or pure THC, but rather are using botanical or botanically derived products. These products typically have varying ratios of THC:CBD but there are few studies that have investigated the optimal ratios of the two primary cannabinoids for treating medical conditions. Most studies that have investigated the interactions between these two cannabinoids have limited their scope to the 1:1 ratio typically found in nabixmols (Sativex^®^) [14,15,16,17]. While our data suggest that a 1:1 ratio of CBD to THC may not be optimal, there is a benefit of including CBD in combination with THC to reduce neuropathic pain, as lower levels of THC produced a reduction in sensitivity when CBD was included, this is especially true for the botanical extracts.

Here, we found that pure CBD alone had little impact on acute pain associated with CIPN. This is in contrast to several other studies that have found that CBD can reduce neuropathic pain in animal models [10,12,19]. An important difference between our study and the previous work is that we are looking at the acute effects of CBD administration as compared to those studies that looked at more prolonged effects of CBD treatment. Interestingly, we did observe that botanically derived CBD mixtures were able to reduce pain in neuropathic animals at a dose where CBD alone was ineffective. While there was a small amount of THC in this extract, the dose administered (0.13 mg/kg) would be too low to account for the observed reduction in pain. A number of the terpenes present in the CBD extract, such as β-caryophyllene and α-humulene, have been found to have anti-nociceptive properties in their own rights [20,21]. Further studies will need to be conducted to determine which other constituents of the CBD extract contribute to the improved pain tolerance observed in neuropathic mice or if CBD contributes at all.

In contrast, we found that THC, when administered at a dose as low at 2 mg/kg, was able to reduce neuropathic pain when combined with CBD, and the level to which pain was reduced was fairly consistent regardless of the ratio of THC:CBD tested (particularly for botanical extracts). Although, in male mice there is a significantly greater reduction in mechanical sensitivity by pure THC when CBD was excluded. These data are consistent with a recent meta-analysis that found that high THC:CBD ratios were better at reducing pain severity across a wide range of conditions, including diabetic neuropathy, in patients [22]. These data are also consistent with reports that CBD is an antagonist at cannabinoid receptors 1 and 2, and can blunt the effects of THC [23,24,25]. Based upon the large number of studies that have reported that nabiximols (mixtures of approximately 1:1 THC:CBD) have antinociceptive properties, it was unexpected that we did not see a larger difference in anti-nociceptive potential with different ratios [14,15,16,17]. Our data suggest that there is little effect of CBD on reducing neuropathic pain, and that most of the reduction in pain can be attributed to THC, although CBD may help to augment the impact of THC. This is true for not only the pure compounds, but also the botanically derived compounds, particularly in male mice (although the highest concentration of THC also was more effective than the highest concentration of CBD in female mice).

The data comparing the botanical extracts to pure compounds also allows us to examine the controversial “entourage” effect, the idea that the whole plant provides an additive benefit over individual pure compounds [26]. While we are not able to directly compare the responses between the animals that received pure compounds and botanical extracts, the animals that received botanical extracts did show a trend towards increased levels of force compared to those receiving pure compounds. Furthermore, animals, particularly males, receiving the botanical extracts exhibited a greater response at much lower levels of THC when in a botanical extract compared to pure THC.

Another potential reason for looking at combinations of THC:CBD is that the addition of CBD may alter the metabolism or prolong the effectiveness of THC at reducing pain. While our current studies did not directly address this, a recent study in a rat model of neuropathic pain actually found that CBD, when co-administered with THC, reduced the therapeutic window in which THC had an effect [11]. This is in contrast to recent data in humans that has reported that co-administration of THC and CBD can prolong the effects of THC [27]. For this reason, studies in mice or rats may not be optimal for examining the interaction of THC and CBD on the metabolism of cannabinoids, but instead such pharmacokinetic studies might best be conducted in human derived microsomes and ultimately patients.

The mechanism by which THC and CBD reduce neuropathic pain in our model was not examined, but will be the work of future studies. One potential mechanism would be that THC is acting through CB1 receptor to reduce the release of neurotransmitters and neuronal excitability [28]. Both CBD and THC are also known agonists of the TRPV1 receptor and have been shown to reduce pain in murine models through this receptor [29]. CBD and THC have also recently been shown to reduce neuropathic pain through both CB1 and CB2 [30]. Our own recent work on cannabigerol (CBG) and neuropathic pain suggests that it is likely to be a complicated interaction between multiple receptors [31], including α2-adrenergic receptors, of which CBG is a known agonist; however, the role of CBD and THC at this receptor are unknown.

## Figures and Tables

**Figure 1 biomedicines-10-02548-f001:**
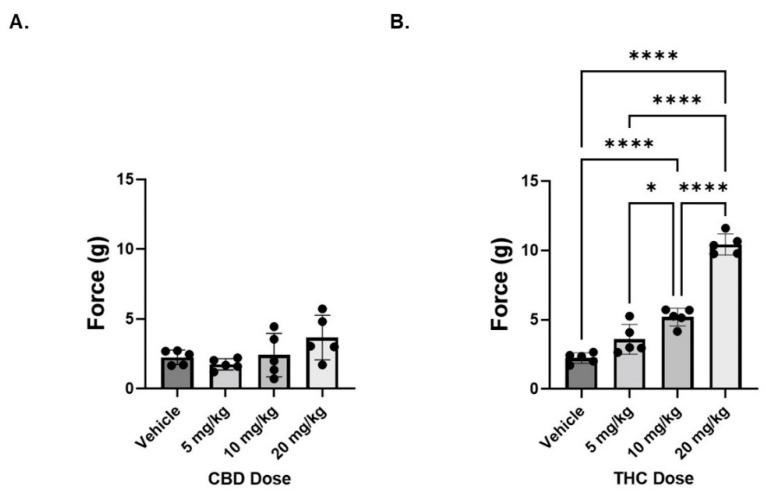
Dose response of cannabinoids on mechanical sensitivity in neuropathic male mice. (**A**) Mice were treated with CBD administered i.p. at 5, 10, or 20 mg/kg 1 h prior to measuring mechanical sensitivity. (**B**) Same as panel A except THC was administered. *n* = 5 mice per group. * *p* < 0.05, **** *p* < 0.001.

**Figure 2 biomedicines-10-02548-f002:**
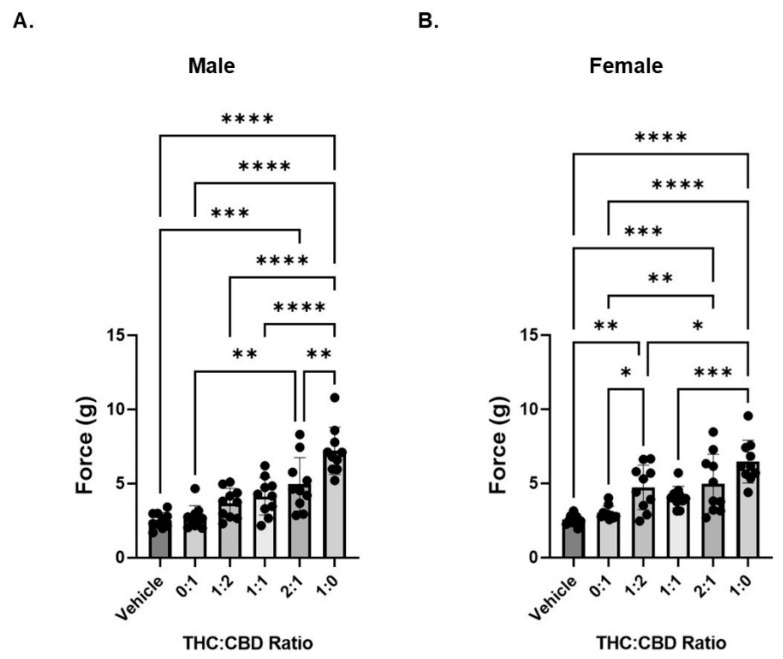
THC reduces mechanical sensitivity in neuropathic mice. (**A**) Neuropathic male mice were treated with 6 mg/kg of total cannabinoid at varying ratios of THC:CBD and mechanical sensitivity was measured by von Frey filament. (**B**) Same as panel A except in female neuropathic mice. *n* = 10 mice per group. * *p* < 0.05, ** *p* < 0.01, *** *p* < 0.005, **** *p* < 0.001.

**Figure 3 biomedicines-10-02548-f003:**
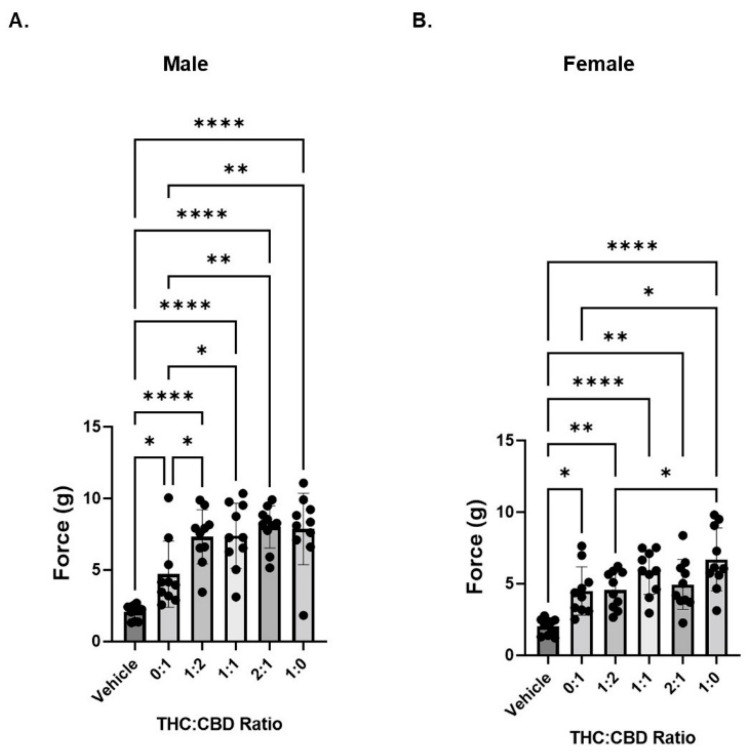
Cannabinoid extracts containing either CBD or THC reduce mechanical sensitivity in neuropathic mice. (**A**) Neuropathic male mice were treated with 6 mg/kg of total cannabinoid at varying ratios of THC:CBD and mechanical sensitivity was measured by von Frey filament. (**B**) Same as panel A except in female neuropathic mice. *n* = 10 mice per group. * *p* < 0.05, ** *p* < 0.01, **** *p* < 0.001.

**Table 1 biomedicines-10-02548-t001:** Dose of cannabinoid administered at each ratio for pure compounds.

Ratio	THC (mg/kg)	CBD (mg/kg)
0:1	0	6
1:2	2	4
1:1	3	3
2:1	4	2
1:0	6	0

**Table 2 biomedicines-10-02548-t002:** Cannabinoid composition of extracts as administered to mice; concentrations are in mg/mL. Cells with no values represent assay results below the level of detection.

Cannabinoid	CBD Extract	THC Extract
CBC		0.03
CBD	1.8	0.005
CBDA		0.003
CBDV	0.03	
CBG		0.067
CBGA		0.002
CBN		0.048
THCA		0.002
Δ^9^-THC	0.04	1.8
**Total Cannabinoid**	**1.87**	**1.957**

**Table 3 biomedicines-10-02548-t003:** Terpene composition of extracts as administered to mice; concentrations are in ppm. Cells with no values represent assay results below the level of detection.

Terpene	CBD Extract	THC Extract
β-Farnesene		3.13
β-Caryophyllene	191.72	20.97
α-Humulene	55.67	4.11
(−) α-Bisabolol	3.61	2.06
β-Myrcene	0.16	2.51
R(+) Limonene	0.17	1.33
Endo-Fenchyl Alcohol	4.33	1.14
Guaiol		1.4
α-Pinene	0.01	3.58
Linalool	2.87	1.54
(−) Caryophyllene Oxide	37.91	
Trans-Nerolidol	3.91	0.61
Valencene		8.2
β-Pinene		1.74
**Total Terpene**	**301.63**	**55.27**
**Total Terpene (mg/mL)**	**0.30**	**0.06**

**Table 4 biomedicines-10-02548-t004:** Dose of cannabinoid administered at each ratio for THC and CBD extracts.

Ratio	THC (mg/kg)	CBD (mg/kg)
0:1	0.13	6
1:2	2.09	4.01
1:1	3.07	3.01
2:1	4.04	2.01
1:0	6	0.02

## Data Availability

Not applicable.

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
