# Peer review of "Combinations of Cannabidiol and Δ^9^-Tetrahydrocannabinol in Reducing Chemotherapeutic Induced Neuropathic Pain"

_biomedicines, 2022, doi:10.3390/biomedicines10102548_

Round 1

Reviewer 1 Report

In this manuscript, the authors use various combinations of THC and CBD, both pure and from plant extracts, to determine efficacy in chemotherapy induced neuropathic pain. They find that while CBD alone has no effect on Von Frey, THC shows good efficacy alone and interacts with CBD. They also demonstrate that using extract-derived CBD/THC, there are stronger anti-allodynic effects, which are attributed to the minor cannabinoids/terpenoids present. 

Generally this is a well-written manuscript that is well-described (especially the extracts/doses) and the conclusions are supported by the data. There are several methodological and interpretation considerations that should be addressed prior to publication, as described below.

1) In the methods, it is stated that mechanical allodynia was measured prior to and after cisplatin treatment. However, in no figure are these pre- or post- cisplatin measurements reported. Since every animal went through this treatment, each group, in each figure, should have their pre-treatment baseline (and possibly post- as well). Otherwise, it’s impossible for the reader to know that cisplatin actually produced an allodynic state. Part of the concern is that often, measurements of 2-2.5 grams is a normal, pain-naïve response in Von Frey (for BL/6 mice).

2) The rationale for 6 mg/kg THC (or total cannabinoid) seems confusing. The authors state that because in Figure 1, 5 mg/kg produced no change while 10 mg/kg did, they decided on 6 mg/kg knowing that it “would serve as a positive control”. How could the authors know that 6 mg/kg would work as a positive control based on negative 5 mg/kg and positive 10 mg/kg data? Logically, if these were the only 2 points, then 10 mg/kg should be the positive control, as it’s the lowest dose that produced an anti-allodynic response. 

Further, why does 6 mg/kg THC produce a stronger response than 10 mg/kg, especially given that 20 mg/kg produces the strongest response? The mean for 10 mg/kg appears to be around 5g, whereas in Figure 2 (1:0, 6 mg/kg THC) males show around 7g and females 6g. The relevance of Figure 1 should be made more clear because currently it is confusing why it is included aside from demonstrating a lack of effect for CBD.

3) The section title 3.2 states “CBD does not influence THC-induced decreases in mechanical hypersensitivity”. However, based on Figure 1 data and Figure 2 data, this must be incorrect. 5 mg/kg THC produces no change in allodynia (Fig 1), but in Fig 2 ratios of 4mg/kg THC:2mg/kg CBD produces a significant change in males, and both 2 THC:4 CBD and 4 THC:2 CBD in females. Thus, because CBD produces no effect on its own, but with lower doses of THC does produce an effect, CBD must then influence THC-induced decreases in allodynia. This is important because it changes the narrative in the discussion with respect to nabiximols etc… In fact the authors have pretty clearly demonstrated that lower doses of THC can be effective when CBD is present. The authors need to amend their narrative or produce additional data that supports their current position. While it certainly appears to be true that no treatment is as good as 6mg/kg THC, that does not negate the fact that lower doses of THC benefit from CBD content. In fact, adding CBD to 6mg/kg THC may provide an even greater benefit. 

4) Were males and/or females used for Figure 1?

5) The conclusions section may benefit from a short discussion on the entourage effect. While controversial, the data from the extracts seems to suggest a benefit of having the “whole plant” compared to THC/CBD only, and this is one of the best, dose-controlled studies to date that has looked at this. 

Author Response

In this manuscript, the authors use various combinations of THC and CBD, both pure and from plant extracts, to determine efficacy in chemotherapy induced neuropathic pain. They find that while CBD alone has no effect on Von Frey, THC shows good efficacy alone and interacts with CBD. They also demonstrate that using extract-derived CBD/THC, there are stronger anti-allodynic effects, which are attributed to the minor cannabinoids/terpenoids present. 

Generally this is a well-written manuscript that is well-described (especially the extracts/doses) and the conclusions are supported by the data. There are several methodological and interpretation considerations that should be addressed prior to publication, as described below.

We thank the reviewer for the kind description of our work and greatly valued the critique of our data and believe it has made our study stronger. 

1) In the methods, it is stated that mechanical allodynia was measured prior to and after cisplatin treatment. However, in no figure are these pre- or post- cisplatin measurements reported. Since every animal went through this treatment, each group, in each figure, should have their pre-treatment baseline (and possibly post- as well). Otherwise, it’s impossible for the reader to know that cisplatin actually produced an allodynic state. Part of the concern is that often, measurements of 2-2.5 grams is a normal, pain-naïve response in Von Frey (for BL/6 mice).

These data have now been added as Supplemental Figures 1-3.  Our pre-cisplatin mice have a von Frey score averaging around 6-7 g and following cisplatin treatment this drops to around 2-3 grams.  

2) The rationale for 6 mg/kg THC (or total cannabinoid) seems confusing. The authors state that because in Figure 1, 5 mg/kg produced no change while 10 mg/kg did, they decided on 6 mg/kg knowing that it “would serve as a positive control”. How could the authors know that 6 mg/kg would work as a positive control based on negative 5 mg/kg and positive 10 mg/kg data? Logically, if these were the only 2 points, then 10 mg/kg should be the positive control, as it’s the lowest dose that produced an anti-allodynic response. 

Further, why does 6 mg/kg THC produce a stronger response than 10 mg/kg, especially given that 20 mg/kg produces the strongest response? The mean for 10 mg/kg appears to be around 5g, whereas in Figure 2 (1:0, 6 mg/kg THC) males show around 7g and females 6g. The relevance of Figure 1 should be made more clear because currently it is confusing why it is included aside from demonstrating a lack of effect for CBD.

We have now added text to better describe why we used 6 mg/kg as the dose of total cannabinoid, and to better define why a dose response was performed.  Additionally, we have added a reference to a prior study that was performed by Diana Sepulveda in her prior lab that found that 6 mg/kg would serve as a positive control. 

It is not entirely clear to us why the 6 mg/kg dose produces a stronger response, although it is important to note that different animals were used in those studies and different animals have different responses.  For this reason, we do not make direct comparisons between the experiments. 

3) The section title 3.2 states “CBD does not influence THC-induced decreases in mechanical hypersensitivity”. However, based on Figure 1 data and Figure 2 data, this must be incorrect. 5 mg/kg THC produces no change in allodynia (Fig 1), but in Fig 2 ratios of 4mg/kg THC:2mg/kg CBD produces a significant change in males, and both 2 THC:4 CBD and 4 THC:2 CBD in females. Thus, because CBD produces no effect on its own, but with lower doses of THC does produce an effect, CBD must then influence THC-induced decreases in allodynia. This is important because it changes the narrative in the discussion with respect to nabiximols etc… In fact the authors have pretty clearly demonstrated that lower doses of THC can be effective when CBD is present. The authors need to amend their narrative or produce additional data that supports their current position. While it certainly appears to be true that no treatment is as good as 6mg/kg THC, that does not negate the fact that lower doses of THC benefit from CBD content. In fact, adding CBD to 6mg/kg THC may provide an even greater benefit. 

We have amended the narrative to reflect these findings.  Our original intent was to examine whether the addition of CBD might actually improve the response to THC (i.e., elicit a greater response to pain) - which was not the case.  We did not look at it from the perspective of CBD lowering the effective dose of THC necessary to reduce pain and thank the reviewer for pointing this out.

4) Were males and/or females used for Figure 1? Only males were used for Figure 1, this was noted in the caption, but is now included in the figure title as well.

5) The conclusions section may benefit from a short discussion on the entourage effect. While controversial, the data from the extracts seems to suggest a benefit of having the “whole plant” compared to THC/CBD only, and this is one of the best, dose-controlled studies to date that has looked at this.

Text has been added to the discussion regarding the entourage effect and how our data fit in with this model.

Reviewer 2 Report

The authors examined the anti-nociceptive effect of CBD and THC in treating neuropathic pain in mice. The different paragraphs, the abstract, the figures and tables, the list of references are very correct. I would advice the authors to explain the mechanism of action of THC and of CBD in the treatment of neuropathic pain. The authors indicated the limitations of this study.

I recommend a minor revision.

Author Response

The authors examined the anti-nociceptive effect of CBD and THC in treating neuropathic pain in mice. The different paragraphs, the abstract, the figures and tables, the list of references are very correct. I would advice the authors to explain the mechanism of action of THC and of CBD in the treatment of neuropathic pain. The authors indicated the limitations of this study.

We thank the reviewer for their kind remarks on our study.  While the mechanism was not examined in our study, we have added text to the discussion on potential mechanisms that may be at work.  The reason the pre-cisplatin graph is the same is that it was the same mice used for both the CBD and THC dose response curves in supplemental figure 1.  As mentioned in the methods to save on the number of animals some experiments were conducted with a 1 week washout period.  The post-cisplatin responses are different because we confirm that the mice are still neuropathic prior to conducting the second treatment.

Round 2

Reviewer 1 Report

The authors have addressed all of my issues and I thank them for their diligence. One small note is that it appears the Pre-Cisplatin force values in Supplemental 1A and B appear to be the same data set. The authors might double check. Otherwise, I have no further issues.